# Tempol Inhibits the Growth of Lung Cancer and Normal Cells through Apoptosis Accompanied by Increased O_2_^•−^ Levels and Glutathione Depletion

**DOI:** 10.3390/molecules27217341

**Published:** 2022-10-28

**Authors:** Woo Hyun Park

**Affiliations:** Department of Physiology, Medical School, Jeonbuk National University, 20 Geonji-ro, Deokjin, Jeonju 54907, Korea; parkwh71@jbnu.ac.kr; Tel.: +82-63-270-3079; Fax: +82-63-274-9892

**Keywords:** tempol, lung cancer cells, human pulmonary fibroblast, cell death, mitochondrial membrane potential, reactive oxygen species, glutathione

## Abstract

Tempol (4-hydroxy-2,2,6,6-tetramethylpiperidine-1-oxyl) is a stable, cell-permeable redox-cycling nitroxide water-soluble superoxide dismutase (SOD) mimetic agent. However, little is known about its cytotoxic effects on lung-related cells. Thus, the present study investigated the effects of Tempol on cell growth and death as well as changes in reactive oxygen species (ROS) and glutathione (GSH) levels in Calu-6 and A549 lung cancer cells, normal lung WI-38 VA-13 cells, and primary pulmonary fibroblast cells. Results showed that Tempol (0.5~4 mM) dose-dependently inhibited the growth of lung cancer and normal cells with an IC_50_ of approximately 1~2 mM at 48 h. Tempol induced apoptosis in lung cells with loss of mitochondrial membrane potential (MMP; ∆Ψm) and activation of caspase-3. There was no significant difference in susceptibility to Tempol between lung cancer and normal cells. Z-VAD, a pan-caspase inhibitor, significantly decreased the number of annexin V-positive cells in Tempol-treated Calu-6, A549, and WI-38 VA-13 cells. A 2 mM concentration of Tempol increased ROS levels, including O_2_^•−^ in A549 and WI-38 VA-13 cells after 48 h, and specifically increased O_2_^•−^ levels in Calu-6 cells. In addition, Tempol increased the number of GSH-depleted cells in Calu-6, A549, and WI-38 VA-13 cells at 48 h. Z-VAD partially downregulated O_2_^•−^ levels and GSH depletion in Tempol-treated these cells. In conclusion, treatment with Tempol inhibited the growth of both lung cancer and normal cells via apoptosis and/or necrosis, which was correlated with increased O_2_^•−^ levels and GSH depletion.

## 1. Introduction

Human lung is a multifaceted organ in sheer architecture. The lung is susceptible to damage by exposure to airborne and bloodborne agents, which increase the risk of lung diseases including fibrosis and cancer [1]. Pulmonary fibroblasts (PF) play a crucial role in repair and restoration following injury to the lung [2]. During pathologic lung repair, scanty or superfluous accumulation of fibroblasts can lead to irregular tissue function and ultimately result in lung disease [2]. Lung cancer is one of the most common lung diseases and one of the most common factors contributing to cancer-related mortality worldwide [3,4]. The two main types of cancer include: small cell lung cancer (SCLC) and non-SCLC (NSCLC) accounting for 10~14% and 85~90% of all lung cancer cases, respectively [3,4]. The paucity of conventional drugs has increased the demand for innovative treatment approaches. Among the chemotherapeutic approaches currently attempted include targeting of apoptosis and necrosis, which are cellular responses to cytotoxic drugs [5,6]. The induction of apoptosis via mitochondrial pathway is associated with the loss of mitochondrial membrane potential (MMP; ∆Ψm) [5]. An indispensable event of the mitochondrial pathway is the efflux of cytochrome *c* from the mitochondria to the cytosol where it consecutively assembles into an apoptosome complex with apoptotic protease-activating factor 1 and caspase-9, subsequently leading to the activation of foremost executioner caspases, especially caspase-3 [6,7]. Another apoptotic pathway related to cell death receptors triggers the activity of caspase-8, followed by caspase-3 [8]. Hence, the targeted inhibition of anti-apoptotic pathways is an attractive option for successful therapeutic strategies against lung cancer.

Reactive oxygen species (ROS) are highly unstable and irritable oxygen moieties including hydrogen peroxide (H_2_O_2_), hydroxyl radicals (^•^OH), and superoxide anions (O_2_^•−^). ROS, especially O_2_^•−^, are constantly generated during mitochondrial oxidative phosphorylation and specifically produced by certain oxidative enzymes [9]. Cells utilize several mechanisms to generate a balance between ROS production and elimination and reduce the harmful effects of ROS. The metabolism of O_2_^•−^ to H_2_O_2_ by superoxide dismutase (SOD) is one such mechanism [10]. H_2_O_2_ is then processed to O_2_ and or H_2_O via catalase or glutathione (GSH) peroxidase [11]. GSH is an important peptide antioxidant, which is known to protect cells from toxic insults [12]. The imbalance of ROS levels due to excess ROS production and/or reduced antioxidant response induces oxidative stress, leading to cell death and tissue inflammation, and results in the progression of many diseases including cancer [13,14].

Tempol (4-hydroxy-2,2,6,6-tetramethylpiperidine-1-oxyl) is a stable synthetic cyclic nitroxide that easily permeates biological membranes, and is conventionally used as a contrast agent in magnetic resonance spectroscopy [15,16]. The compound eliminates a variety of ROS and reduces oxidative process, consequently protecting cells from oxidative damage in vitro and in vivo [15,17]. Several antioxidant mechanisms of Tempol have been proposed to explain these protective effects. It may act as a mimic of SOD and reduce the formation of ^•^OH either by scavenging O_2_^•−^ or by reducing intracellular concentration of Fe(II) [15,17,18]. Usually, such beneficial effects are observed up to micromolar concentrations. For example, 50 to 200 μM Tempol prevents cell death in pyrogallol-treated Calu-6 lung cancer cells and As4.1 juxtaglomerular cells [19,20]. However, its potential clinical application has prompted the need for comprehensive studies investigating their potential toxicity. Substantial evidence suggests that Tempol disrupts ferritin synthesis and Fe metabolism, which induces cell death [21]. In addition, mutagenic effects have been reported in different bacterial strains [22]. The toxicity of Tempol is concentration-dependent and is targeted at cancer cells [23]. It is reported that Tempol at millimolar concentrations induces cell growth inhibition and apoptosis depending on cell types and origins [23,24,25]. In addition, high concentration or prolonged exposure of Tempol increases ROS levels in As4.1 cells [24] and SKOV3 ovarian cancer cells [25,26]. Therefore, in order to elucidate the pro- and anti-oxidant effects of Tempol on cells and tissues, further studies are needed to re-evaluate its biological functions and roles. 

Nonetheless, the molecular mechanisms underlying the cytotoxic and anti-growth effects of Tempol on lung-related cancer and normal cells remain unclear. Therefore, the current study investigated the molecular mechanism underlying the anti-growth effects of Tempol in relation to cell death as well as cellular redox status (ROS and GSH level changes) using A549 and Calu-6 lung cancer cells. The present study also analyzed the cytological effects of Tempol in primary normal human PF (HPF) cells and SV40-transformed normal HPF (WI-38 VA-13 subclone 2RA) cells.

## 2. Materials and Methods

### 2.1. Cell Culture

Human SCLC Calu-6 cells, NSCLC A549 adenocarcinoma cells, and normal lung WI-38 VA-13 subclone 2RA cells were obtained from the American Type Culture Collection (Manassas, VA, USA). Primary normal HPF cells were purchased from PromoCell GmbH (C-12360, Heidelberg, Germany) and used between passages five and ten. These lung cells were stored in a standard humidified incubator at 37 °C with 5% CO_2_ and cultured in RPMI-1640 medium supplemented with 10% fetal bovine serum (FBS; Sigma-Aldrich Co., St. Louis, MO, USA) and 1% penicillin-streptomycin (GIBCO BRL, Grand Island, NY, USA). Cells were grown in 100 mm plastic cell culture dishes (BD Falcon, Franklin Lakes, NJ, USA) and harvested with trypsin-EDTA (GIBCO BRL Waltham, MA, USA).

### 2.2. Reagents

Tempol was obtained from Sigma-Aldrich Co. and dissolved in methanol (Sigma-Aldrich Co.) at 1 M as a stock solution. Z-VAD-FMK (benzyloxycarbonyl-Val-Ala-Asp-fluoromethylketone), a pan-caspase inhibitor, was purchased from R&D Systems, Inc. (Minneapolis, MN, USA) and dissolved in dimethyl sulfoxide (DMSO; Sigma-Aldrich Co.) to generate 10 mM stock solution. DMSO (0.1%) was used as a vehicle control. All stock solutions were wrapped in foil and stored at 4 °C or −20 °C.

### 2.3. Cell Growth Inhibition Assay

The effects of Tempol on the growth of lung cells were determined using 3-(4,5-dimethylthiazol-2-yl)-2,5-diphenyltetrazolium bromide (MTT, Sigma-Aldrich Co.) assays. Briefly, cells were seeded into 96-well microtiter plates (Nunc, Roskilde, Denmark) at a density of 5 × 10^4^ cells/well. After incubation with Tempol at indicated concentrations (0.5~4 mM) for 48 h, 20 μL of MTT solution [2 mg/mL in phosphate-buffered saline (PBS; GIBCO BRL)] was added to each well. Plates were then incubated at 37 °C for 4 h. The medium in each well was removed by pipetting, followed by the addition of 100~200 μL of DMSO to solubilize formazan crystals. Optical density was measured at 570 nm with a microplate reader (Synergy™ 2, BioTekR Instruments Inc. Winooski, VT, USA).

### 2.4. Sub-G1 Cell Analysis

Cells in sub-G1 phase were analyzed using propidium iodide (PI, Sigma-Aldrich Co.; Ex/Em = 488 nm/617 nm) staining as previously described [27]. Briefly, 1 × 10^6^ cells in 60-mm culture dishes (BD Falcon) were incubated with Tempol at indicated concentrations for 48 h. Cells were washed with PBS and then incubated with 10 μg/mL PI and RNase (Sigma-Aldrich Co.) at 37 °C for 30 min. Proportions of cells with sub-G1 DNA content were measured and analyzed with a FACStar flow cytometer (BD Sciences, Franklin Lakes, NJ, USA).

### 2.5. Detection of Apoptosis

Apoptosis was identified via annexin V-fluorescein isothiocyanate staining (FITC, Life Technologies, Carlsbad, CA, USA; Ex/Em = 488/519 nm) as previously described [28]. Briefly, 1 × 10^6^ cells in 60-mm culture dishes (BD Falcon) were preincubated with or without Z-VAD (15 μM) for 1 h and then treated with Tempol at indicated concentrations for 48 h. Cells were washed twice with cold PBS and then suspended in 200 μL of binding buffer (10 mM HEPES/NaOH pH 7.4, 140 mM NaCl, 2.5 mM CaCl_2_) at a density of 5 × 10^5^ cells/mL at 37 °C for 30 min. After adding Annexin V-FITC (2 μL), cells were analyzed with a FACStar flow cytometer (BD Sciences).

### 2.6. Measurement of MMP (ΔΨm)

MMP (ΔΨm) was monitored with rhodamine 123 (Sigma-Aldrich Co.; Ex/Em = 485/535 nm), a cell-permeable fluorescent dye that efficiently moves in mitochondria. Briefly, 1 × 10^6^ cells in 60-mm culture dishes (BD Falcon) were incubated with Tempol at indicated concentrations for 48 h. Cells were washed twice with PBS and incubated with rhodamine 123 (0.1 mg/mL) at a concentration of 5 × 10^5^ cells/mL at 37 °C for 30 min. Depolarization of MMP (∆Ψm) results in loss of rhodamine 123 from the mitochondria and decreases the intracellular fluorescence intensity of this dye [27]. Rhodamine 123 staining intensities were determined using a FACStar flow cytometer. Rhodamine 123-negative (−) cells indicated MMP (∆Ψm) loss in cells.

### 2.7. Quantification of Caspase-3 Activity

The activity of caspase-3 in lung cells was evaluated using caspase-3 Colorimetric Assay Kits (R&D systems, Inc.). In brief, 1 × 10^6^ cells were incubated with 2 mM Tempol for 48 h. Cells were washed with PBS and 4 volumes of lysis buffer (Intron Biotechnology) were added. Samples containing 50 μg of total protein were added to 2X Reaction buffer containing DEVD-pNA for caspase-3 activity in 96-well microtiter plates (Nunc) and incubated at 37 °C for 1 h. Optical density was measured at 405 nm using a microplate reader (Synergy™ 2). Each plate contained multiple wells under a given experimental condition as well as multiple control wells. Caspase-3 activity was expressed in arbitrary units.

### 2.8. Determination of Intracellular ROS and O_2_^•−^ Levels

Intracellular ROS, such as H_2_O_2_, ^•^OH, and ONOO^•^, were measured using a fluorescent probe dye, 2′,7′-dichlorodihydrofluorescein diacetate (H_2_DCFDA, Ex/Em = 495 nm/529 nm; Invitrogen Molecular Probes, Eugene, OR, USA), as previously described [29,30]. H_2_DCFDA is poorly sensitive to O_2_^•−^. However, a fluorogenic probe of dihydroethidium (DHE, Ex/Em = 518 nm/605 nm; Invitrogen Molecular Probes) selectively interacts with O_2_^•−^ among the ROS [29]. In brief, 1 × 10^6^ cells in 60 mm culture dishes (BD Falcon) were pretreated with or without Z-VAD (15 μM) for 1 h, followed by treatment with 2 mM Tempol for 48 h. The cells were then washed in PBS and incubated with 20 µM H_2_DCFDA or DHE at 37 °C for 30 min. The mean DCF and DHE fluorescence values were detected using a FACStar flow cytometer (BD Sciences). The mean DCF and DHE levels are expressed as percentages compared to the control cells.

### 2.9. Detection of Intracellular GSH Levels

Cellular GSH levels were evaluated using a fluorescent probe, 5-chloromethylfluorescein diacetate (CMFDA, Ex/Em = 522 nm/595 nm; Invitrogen Molecular Probes), as previously described [29]. In brief, 1 × 10^6^ cells in 60 mm culture dishes (BD Falcon) were pretreated with Z-VAD (15 μM) for 1 h and then treated with 2 mM Tempol for 48 h. The cells were washed with PBS and incubated with 5 µM CMFDA at 37 °C for 30 min. The mean CMF fluorescence intensity was determined using a FACStar flow cytometer (BD Sciences). CMF negative (−) staining indicated GSH depletion in cells.

### 2.10. Statistical Analysis

The results represent the mean of two or three independent experiments (mean ± SD). The data were analyzed using Instat software (GraphPad Prism 5.0, San Diego, CA, USA). A Student’s *t*-test or one-way analysis of variance with post hoc analysis (ANOVA) using Tukey’s multiple comparison test was used to determine statistical significance, which was defined as a *p*-values of <0.05.

## 3. Results

### 3.1. Effects of Tempol on Lung Cancer and Normal Cell Growth

Effects of Tempol on growth of lung cancer and normal cells were observed using MTT assays. A dose-dependent reduction in cell growth was observed in Calu-6 lung cancer cells, with a half-maximal inhibitory concentration (IC_50_) of ~1 mM following treatment with Tempol for 48 h (Figure 1A). Additionally, 4 mM Tempol appeared to reduce the growth of Calu-6 cells by about 90% (Figure 1A). The growth of A549 lung cancer cells was also reduced by Tempol, with an IC_50_ of 1~2 mM, after 48 h incubation (Figure 1B). Treatment with 2 mM Tempol decreased the growth of A549 cells by about 80% (Figure 1B). The growth of WI-38 VA-13 cells, a normal HPF cell line, was also inhibited by incubation with Tempol for 48 h, with an IC_50_ of 1~2 mM (Figure 1C). Treatment with 2 mM Tempol appeared to reduce the growth by about 85% (Figure 1C). In addition, Tempol inhibited the growth of primary normal HPF cells in a dose-dependent manner, with an IC_50_ of ~1 mM after 48 h incubation (Figure 1D). Treatment with 2 mM Tempol reduced the growth of primary HPF cells by about 90% (Figure 1D).

### 3.2. Effects of Tempol on Cell Death of Lung Cancer and Normal Cell Cells

Effects of Tempol on cell death were evaluated using sub-G1 cells and annexin V-stained cells. DNA flow cytometric analysis indicated that treatment with Tempol at the tested concentrations significantly increased the number of sub-G1 cells in Calu-6 lung cancer cells at 48 h (Figure 2A). Tempol at 2 mM and 4 mM increased the number of these cells by approximately 8% and 28%, respectively (Figure 2A). Treatment with 1~4 mM Tempol increased the number of sub-G1 A549 cells at 48 h, and 2 mM and 4 mM Tempol increased the number of these cells by approximately 22% and 42%, respectively (Figure 2B). Furthermore, exposure to 1~4 mM Tempol significantly increased the number of sub-G1 WI-38 VA-13 cells, and treatment with 2 mM and 4 mM Tempol increased the number of these cells by approximately 23% and 43%, respectively (Figure 2C). In addition, Tempol at the tested concentrations significantly increased the number of sub-G1 cells in primary HPF cells at 48 h, with 2 mM Tempol increasing the cell number by approximately 44% (Figure 2D).

Furthermore, Tempol at the tested concentrations significantly increased the amount of annexin V-positive Calu-6 cells at 48 h, and exposure to 2 mM and 4 mM Tempol increased the number by approximately 18% and 40%, respectively (Figure 3A). Treatment with 2 mM and 4 mM Tempol increased the number of annexin V-positive A549 cells by approximately 38% and 65%, respectively (Figure 3B). Furthermore, Tempol at 1~4 mM significantly increased the amount of annexin V-positive WI-38 VA-13 cells, and 2 mM and 4 mM Tempol increased the cell number by approximately 30% and 70%, respectively (Figure 3C). Tempol at 0.5~2 mM significantly increased the number of sub-G1 cells in primary HPF cells, with 2 mM Tempol increasing the cell number by approximately 75% (Figure 3D).

### 3.3. Effects of Tempol on Mitochondrial Membrane Potential (MMP; ∆Ψm) in Lung Cancer and Normal Cells

Apoptosis or necrosis is closely related to MMP (∆Ψm) loss. Thus, the loss of MMP (∆Ψm) in Tempol-treated lung cells was evaluated using a rhodamine 123 dye. MMP (∆Ψm) loss in Calu-6 cells was significantly induced by Tempol at concentrations of 0.5~4 mM after 48 h (Figure 4A). Exposure to 2 mM and 4 mM Tempol led to the loss of MMP (∆Ψm) in Calu-6 cells by 20 and 65%, respectively (Figure 5A). Treatment with Tempol at concentrations of 2 mM and 4 mM significantly induced the loss of MMP (∆Ψm) in A549 cells by approximately 44% and 92%, respectively (Figure 4B). In addition, 2 and 4 mM Tempol significantly increased the extent of MMP (∆Ψm) loss in WI-38 VA-13 cells by more than 90%, although 1 mM Tempol did not result in a significant loss of MMP (∆Ψm) (Figure 4C). The MMP (∆Ψm) loss in primary HPF cells was significantly induced by Tempol at concentrations of 0.5~2 mM after 48 h, with 2 mM Tempol increasing by more than 85% (Figure 4D).

### 3.4. Effects of Tempol and/or Z-VAD on Caspase-3 Activity and Cell Death in Lung Cancer and Normal Cells

Caspase-3 plays an essential role as an executioner in apoptosis [31]. It was analyzed whether Tempol induces activation of caspase-3 in lung cancer and normal cells. Treatment with 2 mM Tempol increased caspase-3 activities in Calu-6, A549, and Wi-38 VA-13 cells at 48 h (Figure 5A). The effects of Z-VAD-FMK, a pan-caspase inhibitor, on cell death in Tempol-treated lung cells were analyzed at 48 h. Based on previous experiments related to caspase inhibitors [32], Calu-6, A549, and Wi-38 VA-13 cells were pretreated with Z-VAD at a concentration of 15 μM for 1 h before treatment with 2 mM Tempol. Such concentrations of Tempol were used to distinguish changes in cell death. Z-VAD significantly decreased the levels of annexin V-positive cells in Tempol-treated Calu-6, A549, and Wi-38 VA-13 cells (Figure 5B–D). However, Z-VAD partially reduced the amount of annexin V-positive Tempol-treated A549 cells (Figure 5C).

### 3.5. Effects of Tempol and/or Z-VAD on ROS and GSH Levels in Lung Cancer and Normal Cells

The effects of Tempol and/or Z-VAD on ROS and GSH levels in lung cancer and normal cells were investigated at 48 h. The intracellular levels of different ROS in lung cells were analyzed using H_2_DCFDA dye for non-specific ROS levels and DHE dye was used for O_2_^•−^ levels. As shown in Figure 6A, treatment with 2 mM Tempol significantly decreased DCF (ROS) levels in Calu-6 cells. Z-VAD did not alter the DCF (ROS) levels in Tempol-treated and -untreated cells. However, 2 mM Tempol significantly increased DCF (ROS) levels in A549 and Wi-38 VA-13 cells at 48 h (Figure 6B,C). Z-VAD slightly decreased the DCF (ROS) levels in Tempol-treated A549 cells, whereas it increased the DCF (ROS) levels in Tempol-treated Wi-38 VA-13 cells (Figure 6B,C). Intracellular DHE (O_2_^•−^) levels were significantly increased in Tempol-treated Calu-6, A549, and Wi-38 VA-13 cells at 48 h (Figure 6D–F). Z-VAD slightly decreased the DHE (O_2_^•−^) levels in Tempol-treated Calu-6 cells without affecting the DHE (O_2_^•−^) levels in Tempol-treated A549 cells (Figure 6D,E). Z-VAD decreased the DHE (O_2_^•−^) levels significantly in Tempol-treated Wi-38 VA-13 cells (Figure 6F).

Changes in GSH levels were assessed in lung cells using a CMF fluorescent dye. Treatment with 2 mM Tempol significantly increased the number of GSH-depleted cells in A549, Calu-6, and Wi-38 VA-13 cells by approximately 25%, 40%, and 30%, respectively (Figure 7A–C). Z-VAD did not alter the number of GSH-depleted Tempol-treated Calu-6 cells, but slightly decreased the number of GSH-depleted Tempol-treated A549 cells (Figure 7A,B). Moreover, Z-VAD significantly decreased the number of GSH-depleted cells in Tempol-treated Wi-38 VA-13 cells (Figure 7C).

## 4. Discussion

Tempol has been shown to enhance inflammation and oxidative damage in numerous cell and tissue models in vitro and in vivo [15,17], which suggests that its biological properties are beyond that of archetypal antioxidants. Recent studies suggest that Tempol induces apoptosis in many types of cancer cells and decreases tumor growth in immune deficient mice [23,24,25,33]. However, little is known about the cytotoxicology of Tempol in lung-related cells. Thus, the present study focused on elucidating cytotoxicological effects of Tempol on cell growth, cell death, and cellular redox status in lung cancer and normal cells. In this study, it was confirmed that high concentrations of Tempol were toxic to lung-related cells and induced apoptosis, accompanied by increases in O_2_^•−^ levels and GSH depletion. 

Treatment with Tempol dose-dependently decreased the growth of Calu-6 lung cancer cells with an IC_50_ of approximately 1 mM at 48 h and. The growth of A549 lung cancer and WI-38 VA-13 normal cells was also reduced by Tempol, with an IC_50_ of 1 ~ 2 mM. In addition, Tempol inhibited the growth of primary normal HPF cells in a dose-dependent manner, with an IC_50_ of approximately 1 mM after 48 h incubation. Similarly, 1 mM Tempol reduced the growth of As4.1 juxtaglomerular cells nearly 50% at 48 h [24]. In addition, Tempol at 1~5 mM decreased the growth of ovarian [25,26], breast [23], leukemia [34], and prostate cancer cells [33] at 24 ~ 96 h. It was also reported that Tempol efficiently inhibits the growth of neoplastic cells, compared with the growth of normal cells [23]. However, the current results indicate no significant difference in susceptibility to Tempol between lung cancer cells and lung normal cells. Instead, the susceptibility of primary HPF normal cells to Tempol appears to be high. The differences are probably dependent on the origin of cell types. Eukaryotic cell cycle consists of four different phases: G1 phase, S phase, G2 phase and M phase [35]. DNA flow cytometry indicated that Tempol at 0.5~2 mM concentrations induced cellular arrest at the S phase of cell cycle in Calu-6 and A549 lung cancer cells at 48 h and Tempol at 0.5 mM increased the proportion of primary HPF cells at S phase (data now shown). In addition, 1 mM Tempol induced S phase arrest in As4.1 cells [24]. However, Tempol induces a G1 or G2/M phase arrest in breast [23] and prostate cancer cells [33]. The specific phase of cell cycle arrest by Tempol contributes to the inhibition of growth of various cancer cells. 

Tempol dose-dependently increased the percentages of sub-G1 cells in lung cancer and normal cells. In addition, higher concentrations of Tempol significantly increased the amounts of annexin V-positive cells in lung cancer and normal cells. Similar to the results obtained from MTT assays, the primary HPF cells among the tested cells showed the highest number of sub-G1 and annexin V-positive cells after treatment with Tempol at 2 mM. Generally, the numbers of annexin V-positive cells were higher than those of sub-G1 cells among Tempol-treated cells. The activation of caspase-3 is indispensable for apoptosis [31]. The activation of caspase-3 is known to mediate apoptosis of Tempol-treated cancer cells [24,33]. The current results showed that Tempol increased caspase-3 activities in Calu-6, A549, and Wi-38 VA-13 cells. Furthermore, Z-VAD, a pan-caspase inhibitor, significantly decreased the levels of annexin V-positive cells in Tempol-treated Calu-6, A549, and Wi-38 VA-13 cells. By contrast, the effect of Z-VAD was slightly low in A549 cells, implying that cell death in Tempol-treated A549 cells occurred in a caspase-independent (or necrosis) as well as caspase-dependent manner. Therefore, Tempol is likely to induce lung cell death via apoptosis and/or necrosis. Collectively, these results indicate that Tempol inhibits the growth of lung cells via S phase arrest of cell cycle as well as apoptosis and/or necrosis. 

Apoptosis is tightly related to failure of MMP (∆Ψm) [36,37]. Tempol treatment can induce a breakdown in MMP (∆Ψm) in various cancer cells [24,25,34]. Likewise, Tempol efficiently induced the loss of MMP (∆Ψm) in lung cancer and normal cells. Similar to the results of annexin-stained assays, the degree of MMP (∆Ψm) loss in Calu-6 cells treated with Tempol was lower than in A549 cells treated with the same doses of Tempol. For example, 2 mM Tempol increased MMP (ΔΨm) loss in Calu-6 and A549 cells by approximately 20% and 44%, respectively. Moreover, normal lung cells showed a higher loss of MMP (∆Ψm) than cancer cells after treatment with Tempol, especially 2 mM. Such differences might be due to different basal activities of mitochondria and antioxidant enzymes in each cell type. Interestingly, the degree of MMP (∆Ψm) loss in Tempol-treated lung cells was higher than that of annexin V-positive cells in these cells. These results imply that Tempol has an initial effect on mitochondrial membranes of both lung cancer and normal cells before cells undergo apoptosis. 

Tempol is a redox-cycling nitroxide that promotes ROS metabolism via rapid reversible transfer between the nitroxide, hydroxylamine and oxoammonium cation forms [15]. Therefore, Tempol is a potential redox agent that may play a role as a redox agent depending on the cellular concentration [25,38]. It is reported that Tempol-induced cell death is associated with increased cellular ROS levels [24,25,26]. The present study showed that 2 mM Tempol significantly decreased DCF (ROS) levels in Calu-6 cells at 48 h but increased the DCF (ROS) levels in A549 and Wi-38 VA-13 cells. Z-VAD did not significantly alter DCF (ROS) levels in Tempol-treated Calu-6 and A549 cells. Even Z-VAD appeared to increase DCF (ROS) levels in Tempol-treated Wi-38 VA-13 cells. However, in Calu-6, A549, and Wi-38 VA-13 cells treated with 2 mM Tempol, the intracellular DHE (O_2_^•−^) levels were significantly increased at 48 h. Moreover, Z-VAD significantly decreased the DHE (O_2_^•−^) levels in Tempol-treated Wi-38 VA-13 cells and slightly decreased the DHE (O_2_^•−^) levels in Tempol-treated Calu-6 cells. The current results and other reports demonstrate that the toxicity of Tempol is mediated via disruption of mitochondrial function [24,25,34], suggesting that Tempol primarily interferes with mitochondrial function in lung-related cells and then induces oxidative stress by increasing the production of O_2_^•−^, resulting in apoptosis. Tempol-induced death in lung cancer and normal cells was correlated with changes in DHE (O_2_^•−^) levels rather than DCF (ROS) levels. 

A reduction in GSH content occurs in a cascade of events inducing apoptosis and its depletion occurs when the MMP (ΔΨm) is totally disrupted [39]. Our previous studies reported that intracellular GSH content is inversely related to cell death [29,40,41,42,43]. In addition, Tempol increased GSH depletion in HL60 leukemia and HeLa cervical cancer cells [34,44]. Similarly, the present study demonstrated an increased number of GSH-depleted cells in Calu-6, A549, and Wi-38 VA-13 cells treated with 2 mM Tempol at 48 h. In addition, Z-VAD, which exhibited an anti-apoptotic effect, significantly decreased the number of GSH-depleted cells in Tempol-treated Wi-38 VA-13 cells. However, Z-VAD did not alter the number of GSH-depleted Tempol-treated Calu-6 cells and partially reduced the number of GSH-depleted Tempol-treated A549 cells. Although GSH content may play a vital role in Tempol-induced cell death, it is not the only factor that accurately predicts cell death. 

In conclusion, treatment with Tempol not only inhibited the growth of lung cancer and normal cells but also induced the cell death via apoptosis and/or necrosis, which was correlated with increased O_2_^•−^ levels and GSH depletion (Figure 8). No significant difference in susceptibility to Tempol was found between lung cancer and normal cells. The results of this study provide insight into our understanding of the cytotoxic role of Tempol in lung cancer and normal cells in terms of cell growth inhibition, cell death, and redox status.

## Figures and Tables

**Figure 1 molecules-27-07341-f001:**
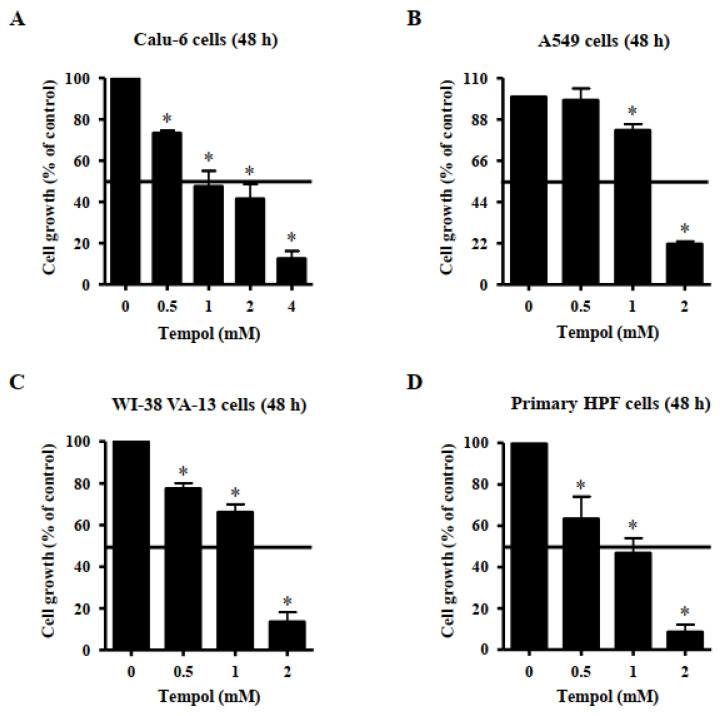
Effects of Tempol on the growth of normal and lung cancer cells. Exponentially growing cells were incubated with Tempol at indicated concentrations for 48 h. Cell growth was evaluated by MTT assays. Graphs show the growth of Calu-6 cancer cells (**A**), A549 cancer cells (**B**), WI-38 VA-13 normal cells (**C**), and primary HPF normal cells (**D**). Student’s *t*-test was used. * *p* < 0.05 compared to Tempol-untreated control cells.

**Figure 2 molecules-27-07341-f002:**
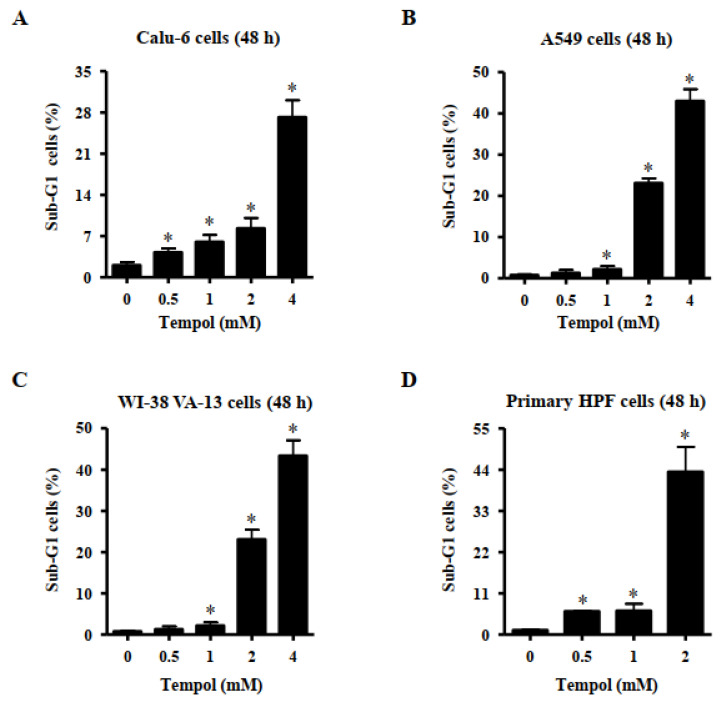
Effects of Tempol on sub-G1 cells of normal and lung cancer cells. Cells in exponential growth phase were incubated with Tempol at indicated concentrations for 48 h. Cells in sub-G1 phase were measured with a FACStar flow cytometer. Graphs show proportions of sub-G1 cells in Calu-6 cells (**A**), A549 cells (**B**), WI-38 VA-13 cells (**C**), and primary HPF cells (**D**). Student’s *t*-test was used. * *p* < 0.05 compared to Tempol-untreated control cells.

**Figure 3 molecules-27-07341-f003:**
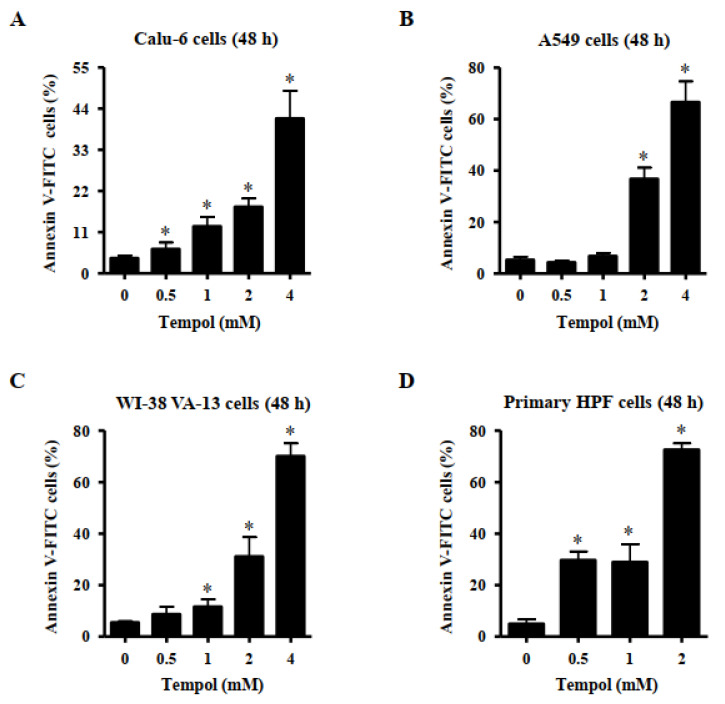
Effects of Tempol on annexin V-positive lung cancer and normal cells. Cells in exponential growth phase were incubated with Tempol at indicated concentrations for 48 h. Annexin V-FITC positive cells were evaluated with a FACStar flow cytometer. Graphs show the proportion of annexin V-positive Calu-6 cells (**A**), A549 cells (**B**), WI-38 VA-13 cells (**C**), and primary HPF cells (**D**). Student’s *t*-test was used. * *p* < 0.05 compared to Tempol-untreated control cells.

**Figure 4 molecules-27-07341-f004:**
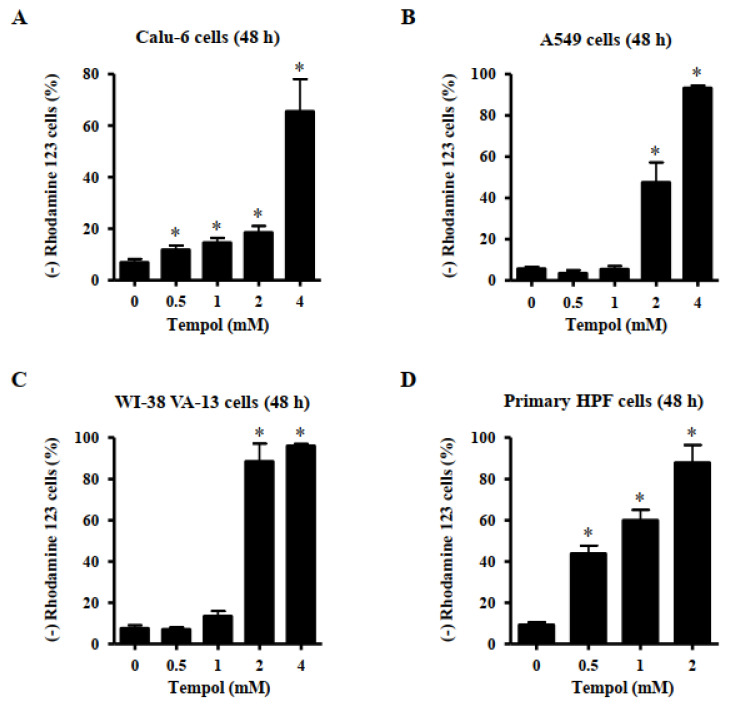
Effects of Tempol on MMP (∆Ψm) levels in lung cancer and normal cells. Exponentially growth of cells incubated with Tempol at indicated concentrations for 48 h. MMP (∆Ψm) in lung cells was measured using a FACStar flow cytometer. Graphs show the proportion of rhodamine 123-negative [MMP (∆Ψm) loss] Calu-6 cells (**A**), A549 cells (**B**), WI-38 VA-13 cells (**C**), and primary HPF cells (**D**). Student’s *t*-test was used. * *p* < 0.05 compared to Tempol-untreated control cells.

**Figure 5 molecules-27-07341-f005:**
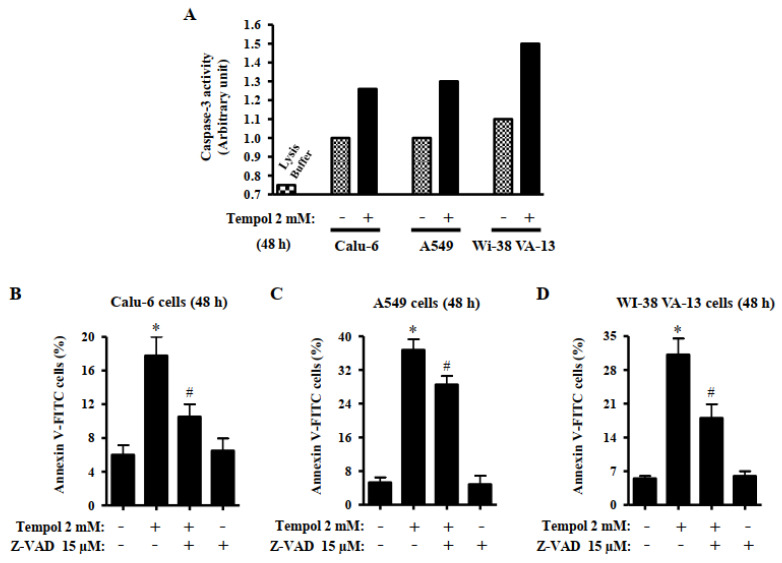
Effects of Tempol and/or Z-VAD on caspase-3 activity and cell death in Calu-6, A549, and WI-38 VA-13 cells. Exponentially growing cells were pretreated with Z-VAD for 1 h and then treated with 2 mM Tempol for 48 h. (**A**): Graph shows the activities of caspase-3 in Calu-6, A549, and Wi-38 VA-13 cells, measured via Colorimetric Assay. (**B**–**D**): Graphs show the proportion of annexin V-positive Calu-6 cells (**B**), A549 cells (**C**), and WI-38 VA-13 cells (**D**), measured with a FACStar flow cytometer. ANOVA test was used. * *p* < 0.05 compared to Tempol-untreated control cells. # *p* < 0.05 compared to cells treated with Tempol only.

**Figure 6 molecules-27-07341-f006:**
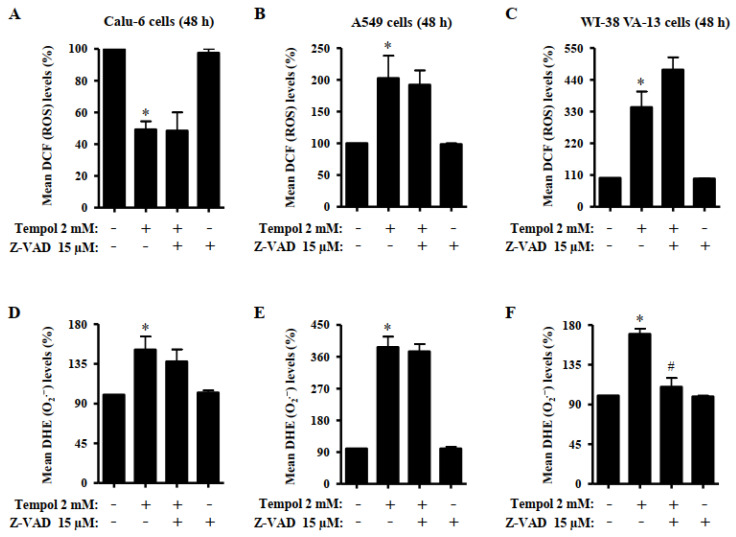
Effects of Tempol and/or Z-VAD on ROS levels in Calu-6, A549, and WI-38 VA-13 cells. Exponentially growing cells were pretreated with Z-VAD for 1 h and then treated with 2 mM Tempol for 48 h. Intracellular DCF (ROS) and DHE (O_2_^•−^) levels in lung cells were measured using a FACStar flow cytometer. (**A**–**C**): The graphs indicate the mean DCF (ROS) levels (%) in Calu-6 (**A**), A549 cells (**B**), and WI-38 VA-13 cells (**C**). (**D**–**F**): The graphs indicate the mean DHE (O_2_^•−^) levels (%) in Calu-6 (**D**), A549 cells (**E**), and WI-38 VA-13 cells (**F**). ANOVA test was used. * *p* < 0.05 compared to Tempol-untreated control cells. # *p* < 0.05 compared to cells treated with Tempol only.

**Figure 7 molecules-27-07341-f007:**
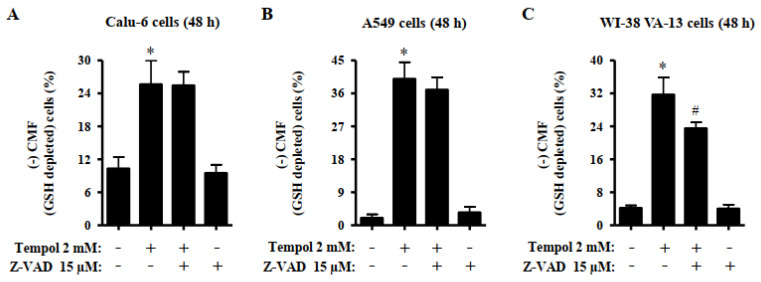
Effects of Tempol and/or Z-VAD on intracellular GSH depletion in Calu-6, A549, and WI-38 VA-13 cells. Exponentially growing cells were pretreated with Z-VAD for 1 h and then treated with 2 mM Tempol for 48 h. The intracellular CMF (GSH) levels in lung cells were measured using a FACStar flow cytometer. The graphs indicate the percentages of (-) CMF (GSH-depleted) Calu-6 cells (**A**), A549 cells (**B**), and WI-38 VA-13 cells (**C**). ANOVA test was used. * *p* < 0.05 compared to Tempol-untreated control cells. # *p* < 0.05 compared to cells treated with Tempol only.

**Figure 8 molecules-27-07341-f008:**
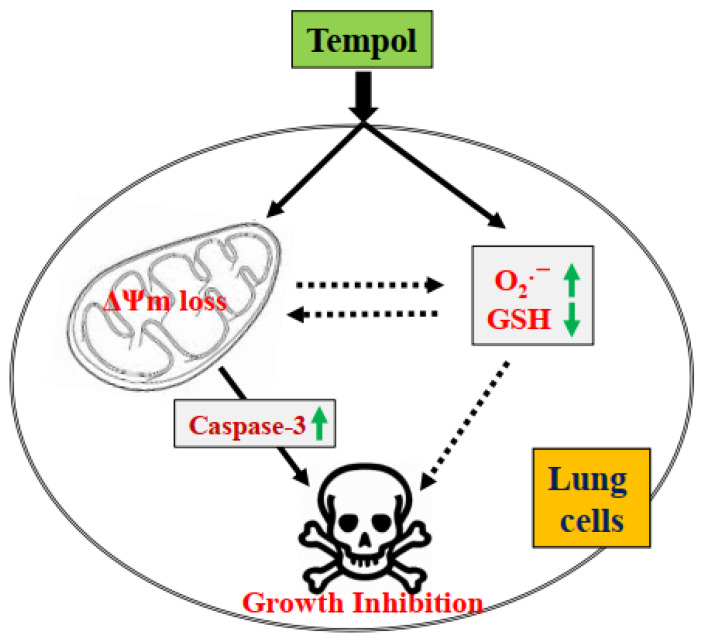
Schematic diagram of Tempol-induced growth inhibition of lung cells.

## Data Availability

Data collected during the present study are available from the corresponding author upon reasonable request.

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
