# Peer review of "Tempol Inhibits the Growth of Lung Cancer and Normal Cells through Apoptosis Accompanied by Increased O2•− Levels and Glutathione Depletion"

_molecules, 2022, doi:10.3390/molecules27217341_

Round 1
Reviewer 1 Report
Tempol is a chemical mimetic of superoxide dismutase activity, which earlier has been shown to have cell protective effects when applied in micromolar concentrations. Park assessed he effects of low micromolar tempol concentration on cancer and normal cells. He demonstrated that in these concentrations tempol suppresses cell growth and induces apoptosis in a caspase-dependent maner. Moreover, micromolar concentrations of tempol drastically decreased mitochondrial membrane potential, increased intracellular ROS level and caused GSH depletion; and these effects did not depend on caspase activity. This is a well done study, its results could be useful when considering the potential pharmaceutic applications of tempol.
Author Response
Thank you for your valuable comments and positive evaluations.
I appreciate Editor and Reviewer for their considerate cooperation.
Reviewer 2 Report
Dear Author,
Both the design and methodology of the study were chosen correctly and prove the presented thesis. The prepared figures are legible and consistent with the description. The text is comprehensible and free of major grammatical or stylistic errors.
However some minor issues occurs and they need to be addressed:
- The manuscript (including references) should be well-checked.
- Were the cells analysed for mycoplasma if so, please indicate the test used.
- In the captions to the figure, I would add information about the statistical test used (for exampe Figure 1-7).
Author Response
Both the design and methodology of the study were chosen correctly and prove the presented thesis. The prepared figures are legible and consistent with the description. The text is comprehensible and free of major grammatical or stylistic errors.
--> Thank you for your valuable comments and positive evaluations.
However some minor issues occurs and they need to be addressed:
- The manuscript (including references) should be well-checked.
--> Thank you very much for your comment. The paper was reviewed by an English editing company, but additional attempts were made to correct the errors in English throughout the manuscript. I also tried to carefully add relevant references to the manuscript.
- Were the cells analysed for mycoplasma if so, please indicate the test used.
--> Thank you for your considerate comment. A long time ago, we tested for mycoplasma to see if the culture medium and culture cells were contaminated. However, we have not detected any mycoplasma. Therefore, we have not recently tested mycoplasma because we have not observed the contamination of cell culture system in our laboratory. Even now, we are doing well in cell culture without any contamination problems.
- In the captions to the figure, I would add information about the statistical test used (for example Figure 1-7).
--> Thank you very much for your comment. Details of the statistical tests used in this manuscript have been described (2.10. Statistical analysis). I agree with your suggestion. Therefore, in the new version of manuscript, Student’s t-test or ANOVA test information was added to the each Figure.
I appreciate Editor and Reviewer for their considerate cooperation.